# DeliGrasp: Inferring Object Properties with LLMs for Adaptive Grasp Policies

**William Xie, Maria Valentini, Jensen Lavering, Nikolaus Correll**[*]

**Abstract:** Large language models (LLMs) can provide rich physical descriptions of most worldly objects, allowing robots to achieve more informed and capable grasping. We leverage LLMs' common sense physical reasoning and code-writing abilities to infer an object's physical characteristics—mass $m$, friction coefficient $\mu$, and spring constant $k$—from a semantic description, and then translate those characteristics into an executable adaptive grasp policy. Using a two-finger gripper with a built-in depth camera that can control its torque by limiting motor current, we demonstrate that LLM-parameterized but first-principles grasp policies outperform both traditional adaptive grasp policies and direct LLM-as-code policies on a custom benchmark of 12 delicate and deformable items including food, produce, toys, and other everyday items, spanning two orders of magnitude in mass and required pick-up force. We then improve property estimation and grasp performance on variable size objects with model finetuning on property-based comparisons and eliciting such comparisons via chain-of-thought prompting. We also demonstrate how compliance feedback from DeliGrasp policies can aid in downstream tasks such as measuring produce ripeness. Our code and videos are available at: https://deligrasp.github.io/

## 1 Introduction

Large language models (LLMs) are able to supervise robot control and learning in manipulation from high-level step-by-step task planning [1, 2, 3] to low-level motion planning [4, 5]. LLMs additionally aid in robot manipulation via understanding a given object's semantic properties and delineating appropriate grasp positions conditioned on those semantic affordances [6, 7, 8].

These works usually assume that the acts of "picking" and "placing" are straightforward tasks. This is not the case for contact-rich manipulation, in which LLM-supervised methods [9, 10, 11] still do not account for force-adaptive tasks like grasping a paper airplane or ripe fruits and vegetables and are prone to damaging such objects. Algorithmic methods for grasping delicate objects [12, 13, 14] require custom hardware and are tested on a limited set of items. LLMs provide an opportunity to leverage their common-sense physical reasoning [15] to produce grasp skills which are both force-adaptive and for the open-world. This is particularly important for semi-structured environments like supermarkets that are subject to a constantly rotating stock or dealing with loose food items such as fruits, vegetables, and pastries that come in a large variety of changing shapes. Yet, LLMs trained on public data are unlikely to contain information on every possible object, in particular when considering specialty domains such as warehouse picking [16] or industrial assembly [17] that often include bespoke parts.

We propose DeliGrasp, an extension of LLM-supervised robot learning to contact-rich manipulation. We posit that LLMs can infer the physical characteristics of gripper-object interactions, including mass, spring constant, and friction. We then formulate an adaptive grasp controller with slip detection derived from the inferred characteristics, endowing LLMs embodied with any force-controllable gripper with adaptive grasp skills for objects spanning a range of weight, size, fragility, and compliance. We also compare DeliGrasp against traditional adaptive grasp algorithms, as well

---

[*1]All authors are with the University of Colorado at Boulder, Boulder, CO. Corresponding email: `wixi6454@colorado.edu`

8th Conference on Robot Learning (CoRL 2024), Munich, Germany.

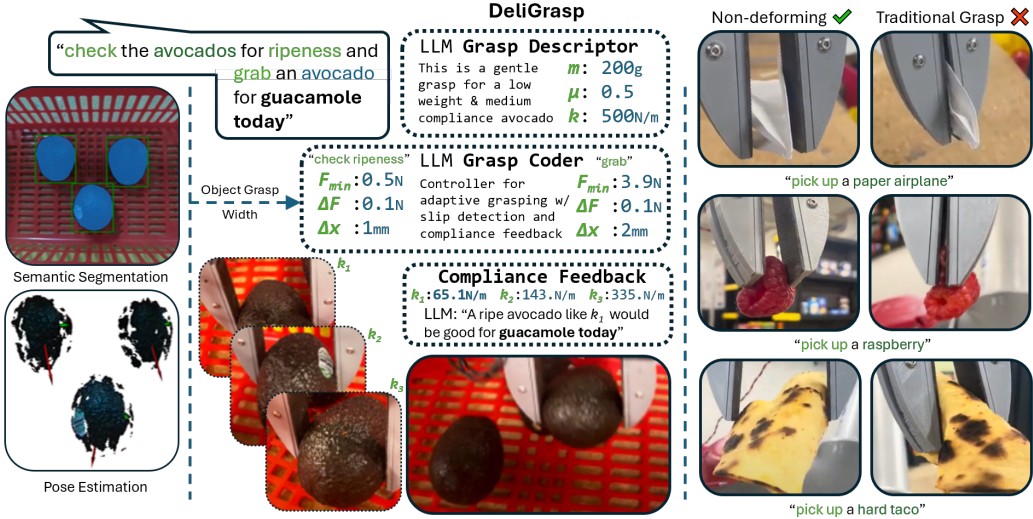

**Figure 1:** Large language models (LLMs) have rich physical knowledge about worldly objects, but cannot directly reason robot grasps for them. Paired with open-world localization and pose estimation (left), our method (middle), queries LLMs for the salient physical characteristics of mass, friction, and compliance as the basis for an adaptive grasp controller. *DeliGrasp* policies successfully grasp delicate and deformable objects (right). These policies also produce compliance feedback as measured spring constants, which we leverage for downstream tasks like picking ripe produce (middle).

as demonstrate how object knowledge can be augmented by finetuning the LLM using property-based comparisons.

We pair DeliGrasp with an open-world localization pipeline which, given an "object description," identifies the object and an initial grasp position. The same "object description" and associated "grasp verb" are the inputs to DeliGrasp, which produces executable Python code controlling a gripper's compliance, force, and aperture as a complete grasp policy for the given description.

By conducting robotic grasping experiments on 12 different objects, we show that DeliGrasp performs successful, minimally-damaging grasps on a custom benchmark of delicate objects which traditional adaptive grasping methods are not capable of. We then improve property estimation and grasp performance using the PhysObjects dataset [11] as well as by eliciting explicit comparisons with chain-of-thought prompting (CoT) [18].

## 2   Related Work

LLMs, equipped with an internet-scale amount of common sense information and logical and physical reasoning [18], are able to comprehend high-level task knowledge, physical context, and robot affordances. For task and motion planning, LLMs can generate navigation and pick-and-place instructions to complete complex and novel tasks [1]. LLM code-writing further augments robot capabilities with closed-loop control [2], new skill generation [3], and translating language to robot parameters for low-level dynamic control [4]. For semantically-afforded grasping, LLMs, in conjunction with other learning methods [8, 19], can identify appropriate grasp locations [6, 7, 20].

For contact-rich manipulation, LLMs have been finetuned on [9, 21] to learn relative object properties such as mass or fragility, but are extensible to low-level control [11]. They have also been prompted with environmental and object properties [10] to parameterize force constraints for robot motion, but not gripper-object interaction. No methods, however, address both the semantics and dynamics requisite for delicate grasping of a broad variety of objects. In comparison, our method utilizes LLMs to estimate object physical properties, which such models are more abundantly pretrained on than code, reward functions, or other niche robot control domains. These estimates are then paired with an underlying algorithmic grasp controller. LLM property estimation, as an approximation of privileged expert knowledge, provides two advantages: simplification of our underlying

adaptive grasp controller relative to classical methods, which rely on in-motion slip detection for adaptive grasping and clarity of the low-level controller such that it is interpretable and predictable.

Grasping delicate objects can be achieved with hardware, such as relying on the compliance of soft grippers, but these end-effectors still require semantic information for control if these grippers must grasp both lightweight and heavy objects [22]. In this paper, we adopt a control algorithm that relies on interaction force measurements from a rigid gripper similar to [12, 13, 23, 24] for minimally deforming grasps and for measuring spring constants [25]. Where these adaptive grasping methods are hardware-specific, ours can be adapted to any force-controllable end-effector, and LLM-estimated object properties reduce controller complexity by approximating privileged information.

## 3  Methods

We source delicate objects for our dataset primarily from supermarkets, kitchens, and food pantries, shown in Figure 2c and describe object mass and object-specific thresholds for unsuccessful, "deforming" grasps in Table 1, shown in Figure 2a. For objects with empty entries in the "Input Phrase" column, we do not modify or add descriptions beyond the name of the object.

**Table 1:** Delicate and Deformable Object Properties

| ID | Object | Width (mm) | Mass (g) | $F_{min}$ (N) | "Object Description" Input to LLM & VLM | Invalidating Deformation |
|---|---|---|---|---|---|---|
| 1 | Paper Airplane | 20 | 0.8 | 0.02 | — | crumples |
| 2 | Cup (empty) | 75 | 3.6 | 0.11 | "empty paper cup" | crumples, creased |
| 3 | Dried Yuba | 30 | 5.5 | 0.16 | "yuba (dried tofu skin)" | cracks, shatters |
| 4 | Raspberry | 20 | 6 | 0.18 | — | juices, torn |
| 5 | Hard Taco | 65 | 15 | 0.44 | "hard-shell tortilla" | cracks, broken |
| 6 | Mandarin | 50 | 56 | 1.65 | — | inelastic deform |
| 7 | Stuffed Toy | 28 | 74 | 2.18 | "tail of a stuffed animal" | inelastic deform |
| 8 | Cup (water) | 75 | 106 | 3.12 | "paper cup filled with water" | spillage |
| 9 | Bag (noodles) | 90 | 191 | 5.62 | "bag containing noodles" | cracks |
| 10 | Avocado | 60 | 204 | 6.00 | — | inelastic deform |
| 11 | Spray Bottle | 50 | 250 | 7.36 | "bottle filled with water" | spillage |
| 12 | Bag (rice) | 80 | 900 | 26.49 | "bag of rice" | N/A |

The evaluated objects span from 0.8 to 900g and soft produce to rigid plastic, and they are commonly grasped in real-world environments. We compute approximate minimum grasping force, $F_{min}$ from measured properties. "Object Description" inputs are paired with a grasp verb, "pick," to DeliGrasp prompts to generate policies. We also qualify what kind of damage or "invalidating deformation" renders a grasp a failure.

Our pipeline takes as input a an "object description", as described in Table 1, and a "grasp verb," which for the delicate object dataset evaluation is uniformly "pick." Our perception method, adapted from [6], takes the "object description" and semantically segments the object from an RGB-D image with OWL-ViT [26] and "Segment Anything" [27]. We crop the corresponding depth image with the generated mask, produce a point cloud object representing the segmented object, and perform Principal Component Analysis to compute a grasp pose that is aligned with the first three principal axes as well as a minimum object grasp width[28], an example is shown in Figure 1, left.

We then prompt the LLM to define and generate grasps with a dual-prompt structure similar to that of Language to Rewards [4], with an initial grasp "descriptor" or "thinker" prompt which produces a structured description, which the subsequent "coder" prompt translates into an executable Python grasp policy that modulates gripper compliance, force, and aperture according to Algorithm 1 (full prompts in Appendix A.6, A.7). To execute grasp policies, we use a UR5 robot arm with the open-source MAGPIE gripper [29] and implement current-draw derived force control, from a range of 0.15 $N$ to 16 $N$ (default output force is 4 $N$). Our experimental setup is shown in Fig. 2bA. We manually score grasp failures according to whether the object slipped or if the damage experienced matches the "invalidating deformation" criterion put forward in 1.

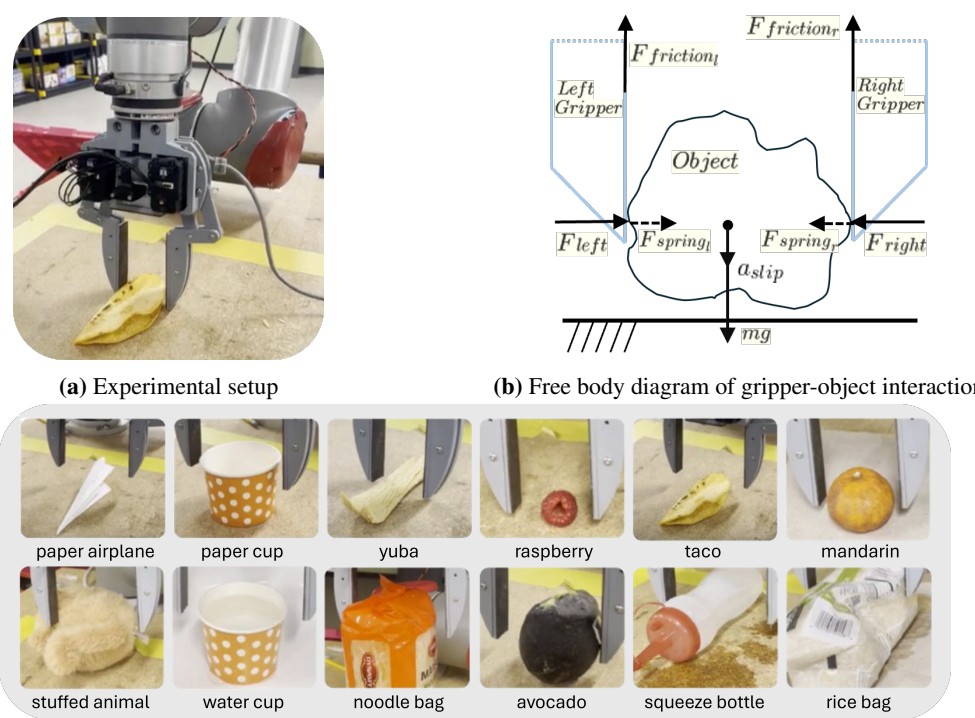

**(a)** Experimental setup      **(b)** Free body diagram of gripper-object interaction

**(c)** Delicate objects dataset

**Figure 3: A.** Our experimental setup with a tabletop UR5 robot arm equipped with the MAGPIE Gripper [29] **B.** Free body diagram describing gripper interactions with an object at rest, adapted from [12] **C.** The delicate objects dataset ranging from 2-900g and various material properties.

### 3.1 Grasp Force Modeling

Fig. 2b shows the interaction between our gripper and an arbitrary object with mass $m$ and spring constant (compliance) $k$. The gripper exerts a composite applied gripper force $F_a = F_{left} + F_{right}$ that leads to a frictional force $F_f = \mu F_a$, where $\mu$ is the Coulomb friction coefficient between the gripper and object [28] that counteracts the force of gravity $mg$ ($g$ is the gravitational constant). For compliant objects, approximated as ideal springs, we can additionally describe the left and right gripper forces $F_{l,r} = F_{spring_{l,r}} = kx_{l,r}$, where $x$ is the compression of the grasped object.

Typically, object slip within a gripper is detected when after the gripper grasps an object at rest, the gripper begins some upward acceleration $a_{lift}$, and an object begins slipping with some downward acceleration $a_{slip}$. Increasing $F_a$ to account for $ma_{lift}$ yields an adaptive minimum applied grasp force $F_{min}$ which prevents slip and is minimally deforming [23]: $F_{min} = \frac{m(g+a_{lift})}{\mu}$ Conversely, when an object is slipping with $a_{slip}$, the applied force $F_{a_{slip}} = \frac{m(g-a_{slip})}{\mu}$ [12].

When $a_{lift,slip}$ are 0 and the gripper and object are at static equilibrium, $F_{min} = \frac{mg}{\mu}$. $F_{min}$ can then be arranged in relation to these quantities, where $m^*, \mu^*$ are LLM-estimated terms approximating ground truth measurements.:

$$F_{a_{slip}} < \frac{mg}{\mu} = F_{min} \leq F_{min,LLM} = \frac{m^*g}{\mu^*} \tag{1}$$

For the delicate objects dataset, we estimate a minimum applied grasping force $F_{min} = \frac{mg}{\mu}$ with a conservative, uniform Coulomb friction coefficient of $\mu = 0.33$, used to compute $F_{min}$ in Table 1.

### 3.2 Delicate Grasping

Mass and friction, $m$ and $\mu$, which determine successful grasp forces, as well as spring constant $k$ form the basis of our force-adaptive algorithm for non-slipping, minimally deforming grasping of

delicate objects. We define a closed-loop force controller: starting from an estimated target aperture that corresponds to the estimated object's width $w$ from RGB-D data, we increase the gripper output force $F_{out}$ and decrease gripper aperture $x$ until sensing a contact force $F_c$ greater than the target $F_{min}$ [24]. To determine the gain terms, $\Delta F_{out}$ and $\Delta x$, i.e. how fast we close the gripper and ratchet up force, the controller uses the agent-determined $k$ and $\Delta x$; we change $F_{out}$ by $c \cdot k\Delta x$, where $c = 0.1$ is a dampening constant. We describe the controller in Algorithm 1.

---

**Algorithm 1** Adaptive Grasping for Minimal Deformation

---

$F_c = \texttt{SetGripper}(x = w, 0)$
**while** $F_c \leq F_{min}$ **do**
$\quad F_{out} \mathrel{+}= c \cdot k\Delta x$
$\quad x \quad \mathrel{-}= \Delta x$
$\quad F_c = \texttt{SetGripper}(x, F_{out})$

---

We query an LLM (GPT-4) for these quantities once to formulate grasp policies. Accurate predictions remove the need for parameter tuning [24] and additional gripper sensors [12, 14]. By default, we instruct the LLM to compute $F_{min} = \frac{mg}{\mu}$ and we do not account for $a_{lift}$, choosing to err closer to object slip than deformation. However, we also provide the LLM with agency to deviate from the default $F_{min}$ depending on the "grasp verb" provided.

On average ($n$=30), generating one DeliGrasp policy with GPT-4 takes 9.98s (3.03s for 1x 'Thinker' prompt query, and 6.65s for 1x 'Coder' prompt query). DeliGrasp policies take on average 4.13s to execute (Appendix A.1). Total grasp time sums to, on average, 14.11s, compared to approximately 0.5s for a force-limited grasp, or 1.7s for a classical adaptive grasp [13].

### 3.3 Improving Property Estimation

To demonstrate how LLM-knowledge can be improved and potentially include bespoke objects, we finetune GPT-3.5-Turbo on 6000 captions of PhysObjects images [11], each describing two objects, their materials, and their relative fragility, deformability, and mass. The dataset captures 276 unique common household objects, with which the delicate objects dataset shares only the mandarin and plastic bottle. We also augment the grasp "descriptor" or "thinker" prompt with CoT prompting [18], paired with finetuned and non-finetuned models, to elicit a series of quantitative comparisons of mass: objects that are 1) lighter and 2) heavier, the mass of a 3) typical object of the category being grasped, and the mass of the 4) user-described object relative to the typical object, thereby producing explicit bounds on the object mass (see Appendix A.8 for greater details on PhysObjects finetuning data format, the full CoT prompt, and CoT query latency).

### 3.4 Classical Adaptive Grasping Baselines

We evaluate DeliGrasp against four classical adaptive grasping baselines, two for each strategy: "In Place" and "In Motion." "In Place" grasping closes the gripper around an object at rest until a measured contact force of 2N or 10N, adapted from [24]. The "In Motion" strategy, adapted from [12, 13], closes around an object at rest until a contact force of 1.5N or 0.5N and then moves upward at approximately 10 mm/s for 5 seconds. During this upward motion, the "In Motion" strategy performs closed-loop adaptive grasping at approximately 80Hz, increasing applied gripper force and velocity proportional to measured slip force. We describe the baselines implementation and differences from DeliGrasp in Appendix A.3.

## 4 Experiments

We benchmark DeliGrasp (*DG*) against five grasp policies: in-place adaptive grasping with 2N and 10N grasp force thresholds, in-motion adaptive grasping with 0.5N and 1.5N initial force, closing the gripper fully or until it is output force limited (F.L.), closing the gripper to the visual width of the object determined by our perception method, and an ablated DeliGrasp-Direct policy (*DG-D*) which directly generates $F_{min}$, $\Delta F_{out}$, and $\Delta x$ without first reasoning about an object's physical properties. We also compare DeliGrasp with PhysObjects finetuning (DG-FT) and with finetuning and chain-of-thought prompting (DG-FT-COT), selecting the model configurations with the highest reduction in mass overestimation (Fig. 5).

We employ each grasp policy 10 times with objects placed randomly within a 30 x 45 cm bounding area on a table, and do not record attempts which receive faulty poses from perception. Quantitative results are shown in Figure 2. Figure 5 shows actual grasps of an empty paper cup and the same paper cup filled with water as well as the LLM-generated code.

**Table 2:** Successful Minimally Deforming Grasps on Delicate and Deformable Objects (*10 trials per object*)

| ID Object | DG | DG-FT | DG-FT CoT | DG-D | In Place 10N | In Place 2N | In Motion 1.5 | In Motion 0.5N | Visual | F.L. |
|---|---|---|---|---|---|---|---|---|---|---|
| 1  Paper Airplane | *10* | *10* | *10* | *10* | 0 | 0 | 2 | 8 | 0 | 0 |
| 2  Cup (empty) | *10* | *10* | *10* | *10* | 0 | 5 | 3 | *10* | *10* | 0 |
| 3  Dried Yuba | 9 | *10* | 8 | 7 | 0 | 3 | 6 | *10* | 3 | 0 |
| 4  Raspberry | 9 | *10* | *10* | 8 | 0 | 0 | 0 | 0 | 0 | 0 |
| 5  Hard Taco | 9 | 6 | 7 | 7 | 0 | 7 | 6 | *10* | 5 | 0 |
| 6  Mandarin | *10* | *10* | *10* | *10* | *10* | *10* | *10* | *10* | *10* | *10* |
| 7  Stuffed Toy | 7 | 6 | 7 | 8 | *10* | *10* | *10* | *10* | 0 | *10* |
| 8  Cup (water) | *10* | *10* | *10* | 8 | 0 | 4 | 7 | 6 | 3 | 4 |
| 9  Bag (noodles) | 7 | 4 | 5 | 4 | 8 | 0 | *9* | 0 | 0 | 5 |
| 10 Avocado | 9 | *10* | 8 | 7 | 8 | 0 | 5 | 2 | 4 | 0 |
| 11 Spray Bottle | *6* | *6* | 5 | 5 | 2 | 0 | 3 | 0 | 0 | 3 |
| 12 Bag (rice) | 0 | 0 | 0 | 0 | 0 | 0 | 0 | 0 | 0 | *2* |
| **Success (%)** | **80.0** | **76.7** | **75.0** | **70.0** | **31.7** | **32.5** | **50.8** | **55.0** | **29.2** | **28.3** |

Number of successful, non-damaging grasps across grasping strategies (columns) on the delicate object evaluation set (rows). First, we test four DeliGrasp variants, where **FT** corresponds to PhysObjects finetuning, and **CoT** to physical reasoning prompting. **DG-D** uses an ablated approach that skips the first-principle model and estimates algorithmic parameters directly. The four variants are comparatively similar in performance, though DG-D deforms objects slightly more than the others due to higher estimated grasping force. We compare to two classical adaptive grasping strategies: **In Place** adaptive grasping until a threshold contact force of 2N or 10N and **In Motion** adaptive grasping with an initial contact force of 1.5 or 10N and then increasing applied force proportional to measured slip. These classical methods are not able to generalize to the full delicate objects dataset and require expert hand-tuning of controller parameters. We include two naive baselines: **Visual**, which closes the gripper to the estimated object width and **F.L.**, which closes the gripper to its maximal force, 16N.

The base DeliGrasp model (80%), finetuned model (76.7%), finetuned model with CoT prompting (75.0%), and ablated DeliGrasp Direct prompt (70.0%) all perform successful, minimally-deforming grasps that the other baselines are not capable of on objects like the paper airplane and raspberry. DeliGrasp dominates hardware-limited (28.3%) and vision-only grasps (29.2%) in 8 out of 12 objects, and is better or on par in 10 out of 12 objects. The "In Place" strategy completes 32.5% (2N) and 31.7% (10N) of grasps successfully. The "In Motion" strategy completes 55.0% (0.5N) and 50.8% (1.5N) of grasps successfully. Per-item performance is shown in Table 2 and full DeliGrasp policy outputs are provided in Appendix A.1.

We look closely at failure modes to better understand the performance of each policy. Given two bounds of failure—*slip* grasps and *deforming* grasps, DeliGrasp and its FT and CoT variants perform the same or better than DeliGrasp Direct in 10 out of 11 cases. These variants overestimate $F_{min}$ terms (individual estimates in Table 7) but primarily perform *slip* failures. However, for all objects *DG-D* estimates even higher $F_{min}$ values and *deforms* objects at a higher rate. DeliGrasp overestimates $F_{min}$ by a factor of 1.95, whereas *DG-D*, which performs additional invalidating deformations, overestimates $F_{min}$ by a factor of 2.55. Since mass, a common quantity, is more accurately estimated by an LLM, resulting adaptive grasp policies parameterized from mass are more robust than policies directly parameterized by an LLM with more erroneous predictions. All policies *slip* at high rates on the stuffed animal (grasped by the tail), spray bottle, bag of noodles, and bag of rice. These objects are non-linearly and/or very compliant and high volume, exerting lever forces on the gripper that likely exceed $F_{min}$.

DeliGrasp on average overestimates object mass by a factor of 2.5, and underestimates mass only for one object (Appendix A.1). Figure 5 shows that mass overestimation can be reduced by employing chain-of-thought reasoning (reduction to factor of 1.82), where the LLM is prompted to explicitly relate its mass estimate to that of other objects, or by fine-tuning GPT-3.5 on the PhysObjects dataset

(reduction to factor of 2.30), or a combination of both methods (reduction to factor of 1.35), with per-item parameter estimate analysis in Appendix A.2. Though the finetuned model and finetuned model with CoT prompting, *DG FT* and *DG FT COT*, performed similarly successful grasps to the base DeliGrasp model and prompt, they produced more accurate, but lower mass, and thus $F$ estimates for the bag of noodles and hard-shelled taco, explaining why resulting policies in 7 have more slip failures. In addition to leading to more faithful force estimates, this also creates an opportunity to extend DeliGrasp to changing inventories or bespoke items in an industrial application.

| Model Type | DG 4 | DG 3.5 | DG CoT 3.5 | DG FT 3.5 | DG FT CoT 3.5 | DG CoT 4 |
|---|---|---|---|---|---|---|
| **Mass Overestimation Factor** | 2.52 | 1.82 | 1.51 | 2.30 | 1.35 | 1.82 |

**(a)** Ratio of mass overestimation ratio for base Model, PhysObject-finetuned models, and models with additional chain-of-thought prompting

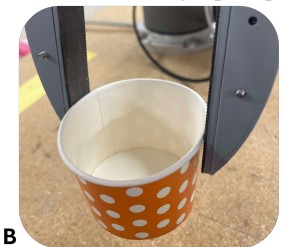
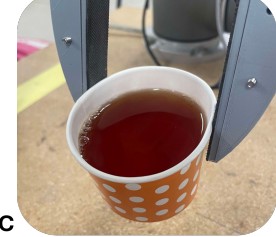

Query: pick up the **empty paper cup**
Estimated properties:
**m: 10g**
**μ: 0.4**
**k: 50.0 N/m**
Controller parameters:
**$F_{min}$: 0.025**
**Δx: 5.0**
**Δf: 0.025**

Query: pick up **paper cup filled with water**
Estimated properties:
**m: 250g**
**μ: 0.6**
**k: 200.0 N/m**
Controller parameters:
**$F_{min}$: 4.0**
**Δx: 2.0**
**Δf: 0.4**

**Figure 5:** (**A**) We compare mass estimates (row) across different LLMs and prompting strategies (columns): denoting that 4/3.5 cprresponds to GPT-4 and 3.5-Turbo, FT to PhysObjects Finetuning, and CoT to physical reasoning prompting. We observe that both finetuning and CoT prompting improve mass estimates, and that the methods together yield the most improved estimates. We show how semantic modifiers like "empty paper cup" (**B**) vs. "paper cup filled with water" (**C**) result in drastic differences on weight and other parameter estimates.

For non-DeliGrasp methods, we observe that qualitatively, the different "In Place" strategies are moderately successful on opposite ends of the evaluation set. The 2N strategy still performs invalidating deformation on the paper plane, empty cup, yuba, raspberry and is not forceful enough for the heavier objects. The 10N strategy deforms lighter objects and delicate heavier objects like the cup of water and the avocado.

The "In Motion" strategies are more performant, but we observe the same performance trends between the 0.5N (more successful on lighter objects) and 1.5N strategies (more successful on heavier objects). We observe that both 0.5N and 1.5N strategies perform slip failures on small and dense objects, such as the avocado or squeeze bottle, due to low surface area of purchase and large measured slip force preventing the controller from adapting in time. Inversely, the controllers are relatively aggressive on lighter or otherwise delicate objects, leading to deformation on the paper plane, water cup, and avocado. Compared to DeliGrasp, LLM common sense reasoning enables a dynamic grasping range in object mass and stiffness.

We note that both classical adaptive grasping methods crush the raspberry. Berry-like objects are low-density, soft, and rough, requiring precise and "delicate" grasping force. The 0.5N "In Motion" strategy applies a low initial force with slight deformation but crushes the raspberry during closed-loop control due to its aggressive gain terms. Categorically, force-limited grasps *deform* and vision-only grasps *slip*. Force-limited grasps succeed only with robust, compliant objects. Vision-only grasps succeed when the objects are relatively stiff and are sensitive to grasp pose.

## 4.1 Grasping Atypical Objects

Since the baseline DeliGrasp prompt policies are performant with overestimated mass, we record grasp performance by querying complex and atypical objects, such as a "larger avocado" (307 g), a wet sponge (92.0 g), a crochet yarn lily flower (26.5 g), and a 10in length of old-growth 2x4 (925 g), and record 10 grasps per configuration (Table 3). We compare across different DeliGrasp configurations—the original DeliGrasp prompt with GPT-4-Turbo, the original prompt with the finetuned GPT-3.5-Turbo model, the CoT prompt with GPT-4-Turbo, and the CoT prompt with the fine-

**Table 3:** Model comparison of mass estimates and grasping success for a typical (143g) and large (306g) avocado

| Model | DG | DG FT | DG CoT | DG FT CoT |
|---|---|---|---|---|
| Larger Avocado | | | | |
| $m$ (g) | 200 | 200 | 250 | 300 |
| Success | 20% | 10% | 80% | 100% |
| Wet Sponge | | | | |
| $m$ (g) | 20 | 50 | 200 | 75 |
| Success | 0% | 30% | 0% | 90% |
| Crochet Yarn Flower | | | | |
| $m$ (g) | 10 | 5 | 5 | 29 |
| Success | 0% | 0% | 0% | 100% |
| Old Growth 2x4 | | | | |
| $m$ (g) | 450 | 450 | 780 | 1000 |
| Success | 0% | 0% | 100% | 100% |

We compare grasp performance from four DeliGrasp configurations, the original DeliGrasp prompt with GPT-4-Turbo (DG), the original prompt with the finetuned GPT-3.5-Turbo model (DG FT), the CoT prompt with GPT-4-Turbo (DG CoT), and the CoT prompt with the finetuned model (DG FT COT), on atypical objects such as a large avocao, a wet sponge, a crochet yarn flower, and old-growth 2x4 lumber. Finetuning, paired with CoT prompting, enables semantically appropriate mass estimates for complex and/or atypical objects.

tuned model. We observe that pairing the fine-tuned model with CoT prompting fully leveraged PhysObjects-like physical reasoning and produced successful grasp policies from an initial descriptive query, due to more accurate initial mass estimates.

The finetuned model with CoT prompting are able to grasp each object, due to more accurate mass estimates resulting from more correct CoT-prompted bounds on object mass. The other methods cause majority slip failures for the larger avocado, the crochet flower, and lumber due to an unchanged mass estimate. For the wet sponge, the other models both overestimate its mass and squeeze the sponge too tightly, or underestimate its mass. The non-finetuned models do not acknowledge the semantic relationship between the provided modifiers ("large," "wet," "crochet yarn," or "old growth") on the mass of a typical object until the user re-queries in the same context with the explicit relationship semantics, such as "the avocado is larger than average, try again." In Appendix A.4, we show how DeliGrasp can also generate atypical grasps conditioned on the verbiage provided (e.g. generating a less forceful grasp for "checking" versus "picking").

## 5   Conclusion

We introduce DeliGrasp, which uses LLMs to 1) infer semantic, common sense physical information which 2) parameterizes adaptive grasp policies effective on a variety of delicate and deformable items that traditional adaptive grasping methods cannot grasp. These inferred $m, \mu, k$ characteristics enable more consistent control than direct estimation of grasping forces (LLM-as-code) and classical adaptive grasping methods. We also improve property estimation accuracy and grasp performance on atypical and complex objects with model finetuning on object property comparisons and chain-of-thought prompting. We then show how DeliGrasp's compliance feedback can be used to measure produce ripeness.

**Limitations and Future Work.** DeliGrasp policies are performant, but dependent on stable and accurate property estimates. For each LLM-estimated quantity in this work, we select the mode of 10 estimates, which can be done offline and reused for simple objects, but not for semantic modifiers or complex and abstract objects. Furthermore, we do not systematically explore the variance of each quantity, or policy robustness to properties besides mass. One interesting line of future work is leveraging iterative observations by mobile robot systems such as humanoids, rather than additional learning methods, to improve system estimates of such properties. We demonstrate this already, as DeliGrasp policies empirically measure $k$ and object width, superseding less-accurate LLM $k$ estimates and perception-based width measurements. We anticipate that perception and physical interactions can produce higher fidelity information to inform manipulation of unknown objects, akin to how humans explore dense, cluttered, and unstructured settings like supermarkets, home kitchens, and other such representative domains. Though finetuning on PhysObjects did not improve grasp performance on our delicate objects dataset, finetuning on this kind of higher fidelity information about complex or out-of-distribution entities may improve grasp performance.

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

# A  Appendix

The MAGPIE gripper has a palm-integrated Intel RealSense D-405 camera. The gripper is powered by two separate Dynamixel AX-12 motors, which allow maximum current control. Translating torque via a four-bar linkage, the gripper force can be controlled and sensed from 0.15 *N* to 16 *N*, and has an aperture range from 106 *mm* fully open to 0 *mm* fully closed (default closure speed of 100 $\frac{mm}{s}$).

## A.1  Full Details of DeliGrasp Performance on Delicate Objects Dataset

Regarding delicate object evaluation, the YCB [30] and RoboCup@Home [31] datasets include some fragile and deformable objects, but there does not exist a dataset focusing on delicate objects with defined failure modes. By combining (1) LLM estimates of object properties to parameterize for robust and damage-free grasping of delicate objects and (2) an algorithmic adaptive grasp controller, our method is able to generalize across many objects and produce uniform grasping feedback which can further inform the controller, LLM, or user.

Computed $\Delta F_{out}$ which are less than 0.01 N are set to 0.01 N. Computed $F_{min}$ values below the force sensing threshold of 0.15 N are set to that threshold when checking for slip or setting output force.

**Table 4:** DeliGrasp Performance on Delicate Objects Dataset (10 trials)

| | | LLM Inferred $m, \mu, k$ and Downstream Terms | | | | | | | Mean Values of Experimentally Produced Terms | | | | | |
|---|---|---|---|---|---|---|---|---|---|---|---|---|---|---|
| ID | Success | $m$ (g) | $\mu$ | $k$ (N/m) | $F_{min}$ (N) $= \frac{mg}{\mu}$ | $F_{err}$ (N) | $\Delta F_{out}$ (N) | $\Delta x$ (mm) | $x_{final}$ (mm) | $x_{goal}$ (mm) | $F_{out}$ (N) | Time (s) | $k_{max}$ (N/m) | $k$ (N/m) |
| 1 | 10 | 5 | 0.5 | 20 | 0.098 | 0.08 | 0.01 | 2 | 11.9 | 7.8 | 0.54 | 4.23 | 332.46 | 184.91 |
| 2 | 10 | 10 | 0.4 | 50 | 0.25 | 0.14 | 0.025 | 5 | 67.83 | 59.5 | 0.5 | 2.94 | 707.93 | 389.76 |
| 3 | 9 | 20 | 0.5 | 200 | 0.39 | 0.23 | 0.04 | 2 | 24.74 | 19.82 | 0.6 | 4.08 | 240.27 | 157.69 |
| 4 | 9 | 5 | 0.8 | 50 | 0.063 | 0.12 | 0.01 | 1 | 15.4 | 5.98 | 0.28 | 7.26 | 97.46 | 60.96 |
| 5 | 9 | 50 | 0.5 | 1000 | 0.98 | 0.54 | 0.2 | 2 | 53.17 | 46.64 | 1.59 | 3.19 | 342.05 | 188.39 |
| 6 | 10 | 150 | 0.8 | 500 | 1.88 | 0.23 | 0.1 | 2 | 45.13 | 39.83 | 2.12 | 3.08 | 236.41 | 151.83 |
| 7 | 9 | 50 | 0.8 | 100 | 0.61 | 1.57 | 0.02 | 2 | 9.79 | 4.19 | 0.97 | 5.29 | 308.73 | 192.77 |
| 8 | 9 | 250 | 0.6 | 200 | 4.08 | 0.96 | 0.04 | 2 | 58.23 | 53.3 | 4.36 | 4.27 | 958.61 | 373.24 |
| 9 | 7 | 500 | 0.4 | 300 | 12.3 | 6.68 | 0.15 | 5 | 71.72 | 61.11 | 12.77 | 2.89 | 7484.7 | 4324.33 |
| 10 | 9 | 200 | 0.5 | 500 | 3.92 | 2.08 | 0.1 | 2 | 50.88 | 46.4 | 4.39 | 3.04 | 729.11 | 298.66 |
| 11 | 6 | 500 | 0.4 | 150 | 12.2 | 4.84 | 0.03 | 2 | 41.94 | 35.08 | 12.6 | 3.64 | 3456.4 | 1383.11 |
| 12 | 0 | 1000 | 0.5 | 200 | 19.6 | 5.9 | 0.1 | 5 | 35 | 27.5 | 20.8 | 5.7 | 13671. | 6192.2 |

We also show the corresponding values for the ablated *DeliGrasp-Direct* policy. We did not record time, experimental $k$ values, or $x_{final}$ in these experiments.

**Table 5:** Ablated DeliGrasp-Direct Performance on Delicate Objects Dataset (10 trials)

| ID | Success | $F_{min}$ (N) | $\Delta F_{out}$ (N) | $\Delta x$ (mm) | $x_{goal}$ (mm) | $F_{out}$ (N) |
|---|---|---|---|---|---|---|
| 1 | 10 | 0.15 | 0.05 | 1 | 4.48 | 0.98 |
| 2 | 10 | 0.5 | 0.2 | 1 | 57.4 | 0.922 |
| 3 | 7 | 0.5 | 0.3 | 2 | 8.6 | 1.14 |
| 4 | 8 | 0.2 | 0.1 | 1 | 6.4 | 0.83 |
| 5 | 7 | 1.5 | 0.2 | 1 | 49.2 | 2.3 |
| 6 | 10 | 1.25 | 0.5 | 5 | 32.4 | 2.01 |
| 7 | 8 | 1.5 | 0.5 | 2 | 3.0 | 3.2 |
| 8 | 8 | 3 | 1 | 5 | 49.6 | 3.18 |
| 9 | 4 | 4 | 1 | 4 | 52.8 | 6 |
| 10 | 7 | 1 | 1 | 5 | 47.2 | 2.2 |
| 11 | 5 | 5 | 2 | 2 | 25.0 | 7.1 |
| 12 | 0 | 8 | 2 | 5 | 105. | 10 |

| ID | Object | Mass (g) | a | a err | b | b err | c | c err | d | d err | e | e err | f | f err | g | g err | h | h err | i | i err | j | j err |
|---|---|---|---|---|---|---|---|---|---|---|---|---|---|---|---|---|---|---|---|---|---|---|
| 1 | paper airplane | 0.8 | 10 | 11.5 | 5 | 6.25 | 5 | 6.25 | 10 | 11.5 | 1 | 1.25 | 5 | 6.25 | 10 | 11.5 | 5 | 5.25 | 10 | 11.5 | 10 | 11.50 |
| 2 | empty cup | 3.6 | 10 | 1.78 | 10 | 2.78 | 10 | 2.78 | 10 | 1.78 | 5 | 1.39 | 5 | 1.39 | 5 | 4.56 | 10 | 1.78 | 20 | 4.56 | 10 | 1.78 |
| 3 | yuba | 5.5 | 20 | 2.64 | 20 | 3.64 | 20 | 3.64 | 10 | 0.82 | 10 | 1.82 | 10 | 1.82 | 10 | 1.82 | 20 | 2.64 | 50 | 8.09 | 20 | 2.64 |
| 4 | raspberry | 6 | 5 | 0.17 | 20 | 3.33 | 5 | 1.17 | 4 | 0.33 | 5 | 1.17 | 4 | 1.33 | 5 | 2.33 | 20 | 2.33 | 20 | 2.33 | 10 | 0.67 |
| 5 | hard-shell tortilla | 15 | 20 | 0.33 | 10 | 1.33 | 50 | 3.33 | 15 | 0.00 | 10 | 1.33 | 30 | 2.00 | 20 | 0.00 | 20 | 0.33 | 15 | 0.00 | 10 | 0.33 |
| 6 | mandarin | 56 | 150 | 1.68 | 150 | 2.68 | 150 | 2.68 | 100 | 0.79 | 50 | 1.11 | 75 | 1.34 | 150 | 1.68 | 150 | 1.68 | 150 | 1.68 | 150 | 1.68 |
| 7 | stuffed animal | 74 | 50 | 0.32 | 50 | 1.32 | 50 | 1.32 | 30 | 0.59 | 20 | 1.73 | 30 | 1.59 | 50 | 0.93 | 50 | 0.32 | 5 | 0.93 | 10 | 0.86 |
| 8 | water cup | 106 | 200 | 0.89 | 200 | 1.89 | 250 | 2.36 | 100 | 0.06 | 150 | 1.42 | 100 | 1.06 | 200 | 1.36 | 200 | 0.89 | 250 | 1.36 | 200 | 0.89 |
| 9 | bag of noodles | 191 | 500 | 1.62 | 200 | 1.05 | 500 | 2.62 | 500 | 1.62 | 200 | 1.05 | 250 | 1.31 | 200 | 1.62 | 500 | 1.62 | 500 | 1.62 | 500 | 1.62 |
| 10 | ripe avocado | 204 | 200 | 0.02 | 200 | 1.02 | 200 | 1.02 | 200 | 0.02 | 200 | 1.02 | 150 | 1.26 | 200 | 0.02 | 200 | 0.02 | 200 | 0.02 | 150 | 0.26 |
| 11 | squeeze bottle | 250 | 300 | 0.20 | 300 | 1.20 | 500 | 2.00 | 300 | 0.20 | 450 | 1.80 | 350 | 1.40 | 400 | 0.40 | 500 | 1.00 | 350 | 0.40 | 200 | 0.20 |
| 12 | bag of rice | 900 | 500 | 0.44 | 1000 | 1.11 | 1000 | 1.11 | 500 | 0.44 | 1000 | 1.11 | 1000 | 1.11 | 500 | 0.44 | 500 | 0.44 | 500 | 0.44 | 500 | 0.44 |
| **Avg Err** | | | | 1.80 | | 2.30 | | 2.52 | | 1.51 | | 1.35 | | 1.82 | | 1.43 | | 1.53 | | 2.74 | | 1.91 |

**Table 6:** Model Mass Estimation: Columns **a** through **j** represent the mass estimates and respective relative error across different model and prompt configurations. The configurations are as follow: **a**: gpt-3.5-turbo, **b**: finetuned gpt-3.5-turbo, **c**: gpt-4-turbo, **d**: gpt-3.5-turbo with CoT prompting, **e**: finetuned gpt-3.5-turbo with CoT prompting, **f**: gpt-4-turbo with CoT prompting, **g**: finetuned gpt-3.5-turbo with 1% training set size, **h**: finetuned gpt-3.5-turbo with 10% training set size, **i**: finetuned gpt-3.5-turbo with 1% training set size and CoT prompting, **j**: finetuned gpt-3.5-turbo with 10% training set size and CoT prompting

We also provide the full per-item breakdown mass estimates from the full set of evaluated models.

## A.2 Per-Item Mass, Friction Coefficient, and Spring Constant Estimate Comparisons for Four DeliGrasp Variants

**Table 7:** DeliGrasp Model Comparisons of Estimated $F_{min}$ and $m$

| Ground Truth | | | | DG | | | | | | DG D | | DG FT | | | | DG FT CoT | | | |
|---|---|---|---|---|---|---|---|---|---|---|---|---|---|---|---|---|---|---|---|
| ID Object Name | F (N) | m (g) | k (N/m) | F (N) | F Err. | m (g) | m Err. | μ | k (N/m) | F (N) | F Err. | F (N) | F Err. | m (g) | m Err. | F (N) | F Err. | m (g) | m Err. |
| 1 Paper Airplane | 0.02 | 0.8 | 184.9 | 0.10 | 4.00 | 5 | 6.25 | 0.5 | 20 | 0.15 | 6.50 | 0.10 | 3.91 | 5 | 6.25 | 0.02 | 0.02 | 1 | 1.25 |
| 2 Cup (empty) | 0.11 | 3.6 | 389.8 | 0.25 | 1.27 | 10 | 2.78 | 0.4 | 50 | 0.5 | 3.55 | 0.25 | 1.23 | 10 | 2.78 | 0.12 | 0.11 | 5 | 1.39 |
| 3 Dried Yuba | 0.16 | 5.5 | 157.7 | 0.39 | 1.44 | 20 | 3.64 | 0.5 | 200 | 0.5 | 2.13 | 0.39 | 1.45 | 20 | 3.64 | 0.20 | 0.23 | 10 | 1.82 |
| 4 Raspberry | 0.18 | 6 | 61.0 | 0.06 | 0.67 | 5 | 1.17 | 0.8 | 50 | 0.2 | 0.11 | 0.25 | 0.36 | 20 | 3.33 | 0.06 | 0.66 | 5 | 1.17 |
| 5 Hard Taco | 0.44 | 15 | 188.4 | 0.98 | 1.23 | 50 | 3.33 | 0.5 | 1000 | 1.5 | 2.41 | 0.20 | 0.55 | 10 | 1.33 | 0.20 | 0.55 | 10 | 1.33 |
| 6 Mandarin | 1.65 | 56 | 151.8 | 1.88 | 0.14 | 150 | 2.68 | 0.8 | 500 | 1.25 | 0.24 | 1.84 | 0.11 | 150 | 2.68 | 0.61 | 0.63 | 50 | 1.11 |
| 7 Stuffed Toy | 2.18 | 74 | 192.8 | 0.61 | 0.72 | 50 | 1.32 | 0.8 | 100 | 1.5 | 0.31 | 0.61 | 0.72 | 50 | 1.32 | 0.25 | 0.89 | 20 | 1.73 |
| 8 Cup (water) | 3.12 | 106 | 373.2 | 4.08 | 0.31 | 250 | 2.36 | 0.6 | 200 | 3 | 0.04 | 3.27 | 0.05 | 200 | 1.89 | 2.45 | 0.21 | 150 | 1.42 |
| 9 Bag (noodles) | 5.62 | 191 | 4324. | 12.3 | 1.19 | 500 | 2.62 | 0.4 | 300 | 4 | 0.29 | 4.91 | 0.13 | 200 | 1.05 | 4.91 | 0.13 | 200 | 1.05 |
| 10 Avocado | 6.00 | 204 | 298.7 | 3.92 | 0.35 | 200 | 1.02 | 0.5 | 500 | 1 | 0.83 | 3.92 | 0.35 | 200 | 1.02 | 3.92 | 0.35 | 200 | 1.02 |
| 11 Spray Bottle | 7.36 | 250 | 1383. | 12.2 | 0.66 | 500 | 2.00 | 0.4 | 150 | 5 | 0.32 | 7.36 | 0.00 | 300 | 1.20 | 11.0 | 0.50 | 450 | 1.80 |
| 12 Bag (rice) | 26.5 | 900 | 6192. | 19.6 | 0.22 | 1000 | 1.11 | 0.5 | 200 | 8 | 0.70 | 19.6 | 0.26 | 1000 | 1.11 | 19.6 | 0.26 | 1000 | 1.11 |

We record LLM estimates for $F$, the minimum grasping force, the relative error $F$ $Err.$, mass $m$, and ratio of mass overestimation $m$ $Err.$ for 5 different DeliGrasp strategies: $DG$ with the default model without finetuning, the ablated $DGD$ which directly estimates $F$, $DG$ $FT$ with the model finetuned on the PhysObjects dataset, and $DG$ $FT$ $CoT$ for the finetuned model with additional chain-of-thought prompting to elicit PhysObjects-like comparisons. We use the same $\mu$ and $k$ values across models to isolate the effect of mass estimates on grasping. Lower mass, and thus $F$ estimates for $DG$ $FT$ $DG$ $FT$ $COT$ for the bag of noodles and hard-shelled taco explain why resulting policies in 7 have more slip failures.

## A.3 Classical Adaptive Grasping Baseline Validation

We first validate these baseline methods on the evaluated objects from their respective studies, spanning a potato chip, tomato, and paper cup (2g to 150g) for "In Place" grasping and an empty styrofoam cup, filled styrofoam cup, empty plastic water bottle, filled plastic water bottle, and filled cereal box (68g to 384g) for "In Motion" grasping. For the "In Motion" strategies, we tune a separate set of applied force gain and velocity gain for the 0.5N and 1.5N contact force strategies. The 0.5N strategy has more aggressive gain terms than the 1.5N strategy, and our current-draw based force sensing used in "In Motion" adaptive grasping is much lower resolution and frequency than dedicated force sensing resistors and slip sensors used in the original baselines.

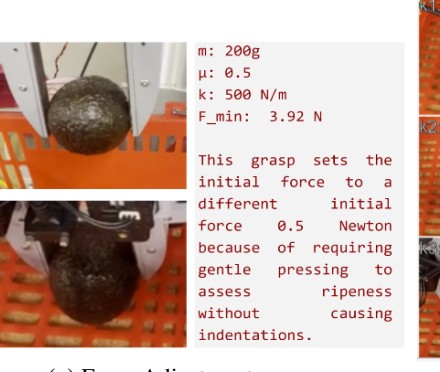
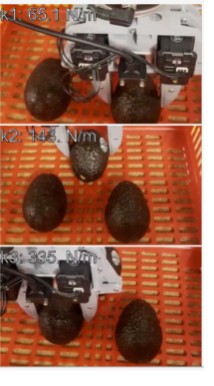
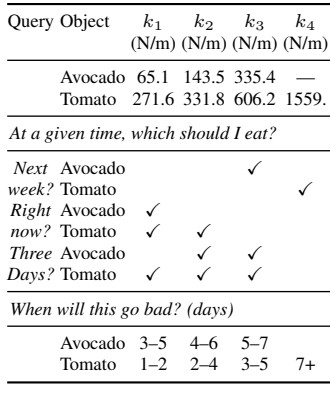

| Query | Object | $k_1$ (N/m) | $k_2$ (N/m) | $k_3$ (N/m) | $k_4$ (N/m) |
|---|---|---|---|---|---|
| | Avocado | 65.1 | 143.5 | 335.4 | — |
| | Tomato | 271.6 | 331.8 | 606.2 | 1559. |
| *At a given time, which should I eat?* | | | | | |
| *Next* | Avocado | | | ✓ | |
| *week?* | Tomato | | | | ✓ |
| *Right* | Avocado | ✓ | | | |
| *now?* | Tomato | ✓ | ✓ | | |
| *Three* | Avocado | | ✓ | ✓ | |
| *Days?* | Tomato | ✓ | ✓ | ✓ | |
| *When will this go bad? (days)* | | | | | |
| | Avocado | 3–5 | 4–6 | 5–7 | |
| | Tomato | 1–2 | 2–4 | 3–5 | 7+ |

| **(a)** Force Adjustment | **(b)** Spring Constant | **(c)** LLM Reasoning |
|---|---|---|

**Figure 6:** DeliGrasp adjusts the grasp force (**A**) for the verb of "checking" the avocado, from the estimated 3.92 N to 0.5 N. Each grasp measures a spring constant $k$ (**B**) without damaging the avocados. Such measurements can be used for downstream LLM-reasoning tasks (**C**) like picking ripe produce or meal planning.

We note that the "In Motion" strategy, in various implementations, requires, in addition to a custom sensor configuration across force, contact, slip, and/or pressure, expert tuning of controller parameters. DeliGrasp requires no sensing beyond the gripper motor(s) current draw and no parameter tuning. Perhaps the even larger obstacle is that such strategies require motion to execute, limiting efficacy on certain geometries and inducing slip uncertainty in a majority of grasps, whereas ours executes entirely in-place and is equally effective across object types.

### A.4 Sensing with DeliGrasp to Pick Ripe Produce

DeliGrasp explicitly measures a spring constant $k$ while performing Algorithm 1: the adaptive grasp controller checks for contact force $F_c$ with each small motion $\Delta x$, and $k = \frac{F_c}{\Delta x}$. These measured $k$ (Appendix A.1) show that DeliGrasp estimates of spring constants are incorrect by a factor of 6.5 (2.5 for only the compliant objects) and, more relevantly, can be utilized as a compliance sensor in tasks such as appraising produce ripeness, a correlate of compliance (Figure 6b and 6c). This sensor data can be used for learning tasks, human collaboration tasks, or for higher-level reasoning tasks such as ripeness checking or appropriateness for specific dishes. We perform this measurement on two assortments of produce: avocados and tomatoes. When receiving the "grasp verb" of "check" or "inspect for ripeness", DeliGrasp bypasses the default minimum grasp force value of $\frac{mg}{\mu}$ and manually sets a lower $F_c$ of 0.5 N for avocados and 0.2 N for tomatoes.

Qualitatively, the relative ordering of the measured $k$ for each item type corresponds with judgments from our own human hand grasps. We query an LLM (GPT-3.5) with the item name and compliance data, and show in Table 6c that it is able to reason over and answer abstract, relative questions about when to eat them. Further qualitative questions such as "how" to eat a specific item "right now" yield dishes appropriate for the ripeness of the produce (Appendix A.5).

### A.5 Ripeness Reasoning with LLMs

1. Asking "how should I eat avocado $k_3$" yields: sliced avocado, grilled avocado, salad toppings, stuffed avocado

2. Asking "how should I eat avocado $k_1$" yields: guacamole, avocado toast, sushi rolls, smoothies

3. Asking "how should I eat tomato $k_4$" yields: sliced for burgers, grilled or roasted, salsa, stuffed

4. Asking "how should I eat tomato $k_1$" yields: caprese salad, gazpacho, pasta sauce

We additionally observe that multimodal models like GPT-4V or OWL-ViT alone cannot reason ripeness over visual images. We query GPT-4V with an image of the same items 5 times and request

a ripeness ranking. For the 4 tomatoes, GPT- 4V produces the same ranking each query, from most ripe to least ripe, of: $[k4, k2, k3, k1]$, reasoning entirely on the "redness" of the tomatoes. For the 3 avocados, it refuses to generate a ranking, citing a fundamental lack of sensory capabilities.

OWL-ViT when directly queried for "the ripest" or "least ripe" produce arbitrarily detects items (sometimes the same item) depending on the angle (and thus lighting, resolution, relative object size) of the captured image.

When we pair the spring constant data with the image in a query to GPT-4V, it ignores the visual characteristics and reasons solely on the spring data.:

Query:

```
I have labeled 3 avocados for you between 0, 1, 2 and provide their
    average spring constants from an initial ripeness check. Spring
    constants are inversely proportional to ripeness--a low spring
    constant means a soft object, and high means stiff. Combining
    this information with the image, please do the following tasks:
1. Estimate the ripeness of a ripe and unripe avocado.
2. Rank the given avocados by ripeness.
E. Using the estimated ripeness range, estimate when each one will go
    bad (avocados are stored in the open at room temperature).

Spring constant data:
Avocado 1: 65 N/m
Avocado 2: 143 N/m
Avocado 3: 335 N/m
```

Response:

```
To perform the tasks, we need to use the given spring constants and
    the visual clues from the image to estimate the ripeness and
    predict when each avocado will spoil. Typically, a ripe avocado
    has a spring constant at the lower end because it is softer,
    while a less ripe or unripe avocado has a higher spring constant
    because it is firmer.

0. Estimate the ripeness of a ripe and unripe avocado:
   - A ripe avocado typically has a low spring constant, indicating
   softness, perhaps similar to Avocado 1, which has the lowest
   spring constant
   - An unripe avocado has a high spring constant, as it's stiffer,
   like Avocado 3, with the highest spring constant

1. Rank the given avocados by ripeness:
   - Avocado 1, with the lowest spring constant range 65, would
   generally be the ripest.
   - Avocado 2, with a spring constant range 143, would be less ripe
   than avo1.
   - Avocado 3, having the highest spring constant range 335, would
   be the least ripe.

So the ripeness rank from ripest to least ripe would be:
Avocado 1 > Avocado 2 > Avocado 3

2. Estimate when each avocado will go bad:
   - Avocado 1, being the ripest, based on its lower spring constant,
   could spoil in about 2-3 days if stored at room temperature.
   - Avocado 2 is less ripe than avo1 and may take around 4-5 days to
   spoil at room temperature.
   - Avocado 3 is the least ripe, with quite a high spring constant
   and could take upwards of a week to spoil, potentially around
   7-10 days at room temperature.

Please note that these estimates are quite rough and actual spoilage
    times can vary due to numerous factors such as specific room
```

```
    temperature, humidity, and the avocado's initial condition. The
    visual clues are limited, but from the image, the physical
    characteristics that usually indicate ripeness (dark skin,
    slightly yielding to gentle pressure) don't provide significant
    additional information beyond the spring constant data.
```

## A.6 DeliGrasp Desciptor Prompt

The descriptor takes as input a "grasp verb", "object description", and "object width." It outputs a structured summary of the object, how it should be grasped, and physical characteristics of the object.

```
Control a robot gripper with force control and contact information.
The gripper's parameters can be adjusted corresponding to the type of
    object that it is trying to grasp as well as the kind of grasp it
    is attempting to perform.
The gripper has a measurable max force of 16N and min force of 0.15N,
    a maximum aperture of 105mm and a minimum aperture of 1mm.

Some grasps may be incomplete, intended for observing force
    information about a given object.
Describe the grasp strategy using the following form:

[start of description]
* This {CHOICE: [is, is not]} a new grasp.
* In accordance with the user instruction, this grasp should be
    [GRASP_DESCRIPTION: <str>].
* This is a {CHOICE: [complete, incomplete]} grasp.
* This grasp {CHOICE: [does, does not]} contain multiple grasps.
* This grasp is for an object with {CHOICE: [high, medium, low]}
    weight.
* The object has an approximate mass of [PNUM: 0.0] grams
* This grasp is for an object with {CHOICE: [high, medium, low]}
    compliance.
* The object has an approximate spring constant of [PNUM: 0.0]
    Newtons per meter.
* The gripper and object have an approximate friction coefficient of
    [PNUM: 0.0]
* This grasp should set the goal aperture to [PNUM: 0.0] mm.
* If the gripper slips, this grasp should close an additional [PNUM:
    0.0] mm.
* If the gripper slips, this grasp should increase the output force
    by [PNUM: 0.0] Newtons.
* [optional] Because of [GRASP_DESCRIPTION: <str>], this grasp sets
    the force to be {CHOICE: [lower, higher]} than the default
    minimum grasp force.
[end of description]

Rules:
1. If you see phrases like {NUM: default_value}, replace the entire
    phrase with a numerical value. If you see {PNUM: default_value},
    replace it with a positive, non-zero numerical value.
2. If you see phrases like {CHOICE: [choice1, choice2, ...]}, it
    means you should replace the entire phrase with one of the
    choices listed. Be sure to replace all of them. If you are not
    sure about the value, just use your best judgement.
3. If you see phrases like [GRASP_DESCRIPTION: default_value], use
    information from the user instruction to provide a description of
    the grasp or the object to be grasped, including mentioned
    physical characteristics or features.
4. Using information from the user instruction about the object and
    the grasp description, set the initial grasp force either to this
    default value or an appropriate value.
```

```
5. If you deviate from the default force value, explain your
   reasoning using the optional bullet points. It is not common to
   deviate from the default value.
6. Using knowledge of the object and how compliant it is, estimate
   the spring constant of the object. This can range broadly from 20
   N/m for a very soft object to 2000 N/m for a very stiff object.
7. Using knowledge of the object and the grasp description, if the
   grasp slips, first estimate an appropriate increase to the
   aperture closure, and then the gripper output force.
8. The increase in gripper output force the maximum value of (0.05 N,
   or the product of the estimated aperture closure, the spring
   constant of the object, and a damping constant 0.1:
   (k*additional_closure*0.0001)).
9. Provide the full description of the grasp plan, even if you may
   only need to change a few lines. Always start the description
   with [start of description] and end it with [end of description].
10. Do not add additional descriptions not shown above. Only use the
    bullet points given in the template.
11. Make sure to give the full description. Do not skip points if
    they are not optional.
"""
```

## A.7 DeliGrasp Coder Prompt

The coder takes as input the structured summary of the grasp and inferred characteristics and generates an grasp policy which implements the adaptive grasping algorithm 1 and compliance sensing.

```
We have a description of a gripper's motion and force sensing and we
    want you to turn that into the corresponding program with
    following class functions of the gripper:
The gripper has a measurable max force of 16N and min force of 0.15N,
    a maximum aperture of 105mm and a minimum aperture of 1mm.

'''
def get_aperture(finger='both')
'''
finger: which finger to get the aperture in mm, of, either 'left',
    'right', or 'both'. If 'left' or 'right', returns aperture, or
    distance, from finger to center. If 'both', returns aperture
    between fingers.

'''
def set_goal_aperture(aperture, finger='both', record_load=False)
'''
aperture: the aperture to set the finger(s) to (in mm)
finger: which finger to set the aperture in mm, of, either 'left',
    'right', or 'both'.
record_load: whether to record the load at the goal aperture. If
    true, will return array of (pos, load) tuples
This function will move the finger(s) to the specified goal aperture,
    and is used to close and open the gripper.
Returns a position-load data array of shape (2, n) --> [[positions],
    [loads]], average force, and max force after the motion.

'''
def set_compliance(margin, flexibility, finger='both')
'''
margin: the allowable error between the goal and present position (in
    mm)
flexibility: the compliance slope of motor torque (value 0-7, higher
    is more flexible) until it reaches the compliance margin
finger: which finger to set compliance for, either 'left', 'right',
    or 'both'
```

```
'''
def set_force(force, finger='both')
'''
force: the maximum force the finger is allowed to apply at contact
    with an object(in N), ranging from (0.1 to 16 N)
finger: which finger to set compliance for, either 'left', 'right',
    or 'both'

'''
def deligrasp(goal_aperture, initial_force, additional_closure,
    additional_force, complete_grasp)
'''
goal_aperture: the goal aperture to grasp the object (in mm)
initial_force: the initial force to apply to the object (in N)
additional_closure: the additional aperture to close if the gripper
    slips (in mm)
additional_force: the additional force to apply if the gripper slips
    (in N)
complete_grasp: whether the grasp is complete or incomplete (True or
    False)
This function will close the gripper to the goal aperture, apply the
    initial force, and adjust the force if the gripper slips. If the
    grasp is incomplete, the gripper will open after the slip check.

'''
def poke(direction, speed, aperture)
'''
direction: 'left' or 'right' to poke the left or right finger
speed of finger in m/s
aperture: distance to poke in mm (of one finger, not both)

Example answer code:
'''
from magpie.gripper import Gripper # must import the gripper class
G = Gripper()
import numpy as np  # import numpy because we are using it below

goal_aperture = {PNUM: goal_aperture}
complete_grasp = {CHOICE: [True, False]}
# Initial force. Convert mass (g) to (kg). The default value of
    object weight / friction coefficient.
initial_force = {PNUM: {CHOICE: [({PNUM: mass} * 9.81) / ({PNUM: mu}
    * 1000), {PNUM: different_inital_force}] }}}
additional_closure = {PNUM: additional_closure}
# Additional force increase. The default value is the product of the
    object spring constant and the additional_closure, with a
    dampening constant 0.1.
additional_force = np.max([0.01, additional_closure * {PNUM:
    spring_constant} * 0.0001])

G.set_goal_aperture(goal_aperture + additional_closure * 2,
    finger='both', record_load=False)
G.set_compliance(1, 3, finger='both')
G.set_force(initial_force, 'both')

G.deligrasp(goal_aperture, initial_force, additional_closure,
    additional_force, complete=complete_grasp, debug=True)
'''

Remember:
1. Always format the code in code blocks. In your response all five
    functions above: get_aperture, set_goal_aperture, set_compliance,
    set_force, check_slip should be used.
```

```
   2. Do not invent new functions or classes. The only allowed functions
      you can call are the ones listed above. Do not leave
      unimplemented code blocks in your response.
   3. The only allowed library is numpy. Do not import or use any other
      library. If you use np, be sure to import numpy.
   4. If you are not sure what value to use, just use your best judge.
      Do not use None for anything.
   5. If you see phrases like [REASONING], replace the entire phrase
      with a code comment explaining the grasp strategy and its
      relation to the following gripper commands.
   6. If you see phrases like [PREDICTION], replace the entire phrase
      with a prediction of the gripper's state after the following
      gripper commands are executed.
   7. If you see phrases like {PNUM: default_value}, replace the value
      with the corresponding value from the grasp description.
   8. If you see phrases like {CHOICE: [choice1, choice2, ...]}, it
      means you should replace the entire phrase with one of the
      choices listed. Be sure to replace all of them. If you are not
      sure about the value, just use your best judgement.
   9. Remember to import the gripper class and create a Gripper at the
      beginning of your code.
   10. Remember to take into account instructions to use different
       forces than the default values, and to explain your reasoning in
       the code.
"""
```

## A.8 PhysObjects Finetuning Data Format, CoT Prompt, and CoT Latency Evaluation

Each text caption is adapted to a pair-wise relative Q&A conversations across these physical concepts concepts, following the structure: *Q*: Which {has more mass, is more fragile, is more deformable}, $material_1$ $object_1$ or $material_2$ $object_2$? and *A*: A $material$ $object$ {has more mass, is more fragile, is more deformable}.

When evaluating mass estimates from finetuned and/or CoT-prompted models for atypical objects, we query the model 10 times for the mode of estimates (minimum 5) as mass estimates are more varied. We execute these queries in parallel, which is a little bit slower than querying an LLM just once, as the final requests are rate-limited and slower to complete, taking on average 3.73s ($n$=30).

We add three new mandatory bullet points to the description, starting after the description of whether the grasp contains multiple grasps. These three bullets elicit CoT thinking and leverage PhysObjects finetuning by requesting explicit mass comparisons between heavier, lighter objects and a typical vs. the user-described object to be grasped. We modify rule 3, add a rule for an example object (rule 4) to further enforce CoT and leverage PhysObjects finetuning.

```
Control a robot gripper with force control and contact information.
The gripper's parameters can be adjusted corresponding to the type of
    object that it is trying to grasp as well as the kind of grasp it
    is attempting to perform.
The gripper has a measurable max force of 16N and min force of 0.15N,
    a maximum aperture of 105mm and a minimum aperture of 1mm.

Some grasps may be incomplete, intended for observing force
    information about a given object.
Describe the grasp strategy using the following form:

[start of description]
* This {CHOICE: [is, is not]} a new grasp.
* In accordance with the user instruction, this grasp should be
    [GRASP_DESCRIPTION: <str>].
* This is a {CHOICE: [complete, incomplete]} grasp.
* This grasp {CHOICE: [does, does not]} contain multiple grasps.
```

```
* This object has more mass than [example object:  <str>], with mass
    of [PNUM: 0.0] g, and less mass than [example object:  <str>],
    with mass of [PNUM: 0.0] g
* Typically, this object's mass is approximately [PNUM: 0.0] g, which
    is between these two masses.
* Because the user specified that [OBJECT_DESCRIPTION: <str>],
    compared to typical, this object has a {CHOICE: [greater, lesser,
    similar]} mass of [PNUM: 0.0] grams.
* This grasp is for an object with {CHOICE: [high, medium, low]}
    compliance.
* The object has an approximate spring constant of [PNUM: 0.0]
    Newtons per meter.
* The gripper and object have an approximate friction coefficient of
    [PNUM: 0.0]
* This grasp should set the goal aperture to [PNUM: 0.0] mm.
* If the gripper slips, this grasp should close an additional [PNUM:
    0.0] mm.
* If the gripper slips, this grasp should increase the output force
    by [PNUM: 0.0] Newtons.
* [optional] Because of [GRASP_DESCRIPTION: <str>], this grasp sets
    the force to be {CHOICE: [lower, higher]} than the default
    minimum grasp force.
[end of description]

Rules:

1. If you see phrases like {NUM: default_value}, replace the entire
    phrase with a numerical value. If you see {PNUM: default_value},
    replace it with a positive, non-zero numerical value.
2. If you see phrases like {CHOICE: [choice1, choice2, ...]}, it
    means you should replace the entire phrase with one of the
    choices listed. Be sure to replace all of them. If you are not
    sure about the value, just use your best judgement.
3. If you see phrases like [GRASP_DESCRIPTION: default_value] or
    [OBJECT_DESCRIPTION: default_value], use information from the
    user instruction to provide a description of the grasp or the
    object to be grasped, including mentioned physical
    characteristics or features.
4. If you see phrases like [example object: <str>], replace the
    entire phrase with an appropriate example object that is similar
    to the object to be grasped.
5. Using information from the user instruction about the object and
    the grasp description, set the initial grasp force either to this
    default value or an appropriate value.
6. If you deviate from the default force value, explain your
    reasoning using the optional bullet points. It is not common to
    deviate from the default value.
7. Using knowledge of the object and how compliant it is, estimate
    the spring constant of the object. This can range broadly from 20
    N/m for a very soft object to 2000 N/m for a very stiff object.
8. Using knowledge of the object and the grasp description, if the
    grasp slips, first estimate an appropriate increase to the
    aperture closure, and then the gripper output force.
9. The increase in gripper output force the maximum value of (0.05 N,
    or the product of the estimated aperture closure, the spring
    constant of the object, and a damping constant 0.1:
    (k*additional_closure*0.0001)).
10. Provide the full description of the grasp plan, even if you may
    only need to change a few lines. Always start the description
    with [start of description] and end it with [end of description].
11. Do not add additional descriptions not shown above. Only use the
    bullet points given in the template.
12. Make sure to give the full description. Do not skip points if
    they are not optional.
"""
```

