# OpenReview forum: "DeliGrasp: Inferring Object Properties with LLMs for Adaptive Grasp Policies"
_robot-learning.org/CoRL/2024/Conference — CoRL 2024_

### Official Review · Reviewer_uYAW · 2024-07-20
**Review of paper 523**

**Originality:** 3
**Technical Quality:** 3
**Clarity Of Presentation:** 2
**Potential Impact:** 3
**Recommendation:** 4
**Confidence:** 3

**Review:**

Strengths:
- Unlike prior work, DeliGrasp leverages the generalization abilities of LLMs without compromising the robustness of a classical adaptive grasp controller.
- The authors demonstrate that this approach can further improve with fine-tuning
- An open question in robotic manipulation is how best to integrate large language models and this paper presents a timely proof-of-concept

Weaknesses:
- Is there a stronger baseline that force limited? E.g., is there a way to develop a compliant grasp as a baseline?
- The clarity of the paper could be improved:
    - Methods section is too detailed. E.g., the paragraph starting on line 94 could be moved to the appendix.
    - It's difficult to understand the purpose of table 2. I would recommend moving it to the appendix, summarizing over object, or adding more analysis of the table to the text. Listing the object id instead of the object name only makes it more difficult to understand the table.
    - Section 4.2 comes as a bit of a surprise. It doesn't advance the main point (that surprisingly LLMs are quite capable of estimating manipulation relevant object properties), so I suggest moving to appendix.
    - It's unclear where the fine-tuned vs not fine-tuned models are used in the experimental section. It would be useful to see a direct comparison between these two variants.

**Quality Of The Limitations Section:**

3

**Questions For Rebuttal:**

- How is success scored in Figure 3? Are the grasps manually labeled as causing damage vs not causing damage?
- Are the deli grasp results in section 4 all using the fine-tuned model?

**Robotics Focus:**

4

**Summary Of Paper:**

The authors propose to use an LLM to predict the mass and friction of an object from a high-level object description. These values parameterize an adaptive grasp controller to pick up the object. The authors further vet the potential of LLM-generated parameters by fine-tuning on a dataset of captioned object images.

**Summary Of Recommendation:**

Overall I recommend acceptance because I think the analysis is interesting, but I think it's important that the authors improve the clarity in the rebuttal period.

---

### Official Review · Reviewer_M14h · 2024-07-21

**Originality:** 3
**Technical Quality:** 2
**Clarity Of Presentation:** 3
**Potential Impact:** 2
**Recommendation:** 3
**Confidence:** 5

**Review:**

### Strengths:
* The proposed method is thoroughly evaluated on a wide range of objects, showcasing its versatility across different properties.
* The methodology is straightforward and easy to implement.

### Weaknesses:
* Novelty: The core features of the study, specifically the use of LLMs for common sense reasoning and code writing, do not introduce new knowledge to the field of robotics. Previous works have already demonstrated these aspects, suggesting that the current study’s contributions may be incremental. While the integration of these features into robotic grasping is noted, the application itself does not substantially advance our understanding or implementation of LLMs in robotics.
* Evaluation: The absence of a comparison with established classical adaptive grasping methods is a significant oversight. For a comprehensive evaluation of the proposed DeliGrasp method, it is essential to benchmark its performance against existing techniques. This comparison would provide critical insights into the advantages offered by LLMs in this context.

Misc:
* The poor image quality in Figure 4 hampers clarity and understanding of the content.
* Incorrect formatting of double quotes in many places (e.g., Table 1).

**Quality Of The Limitations Section:**

2

**Questions For Rebuttal:**

* Considering that DeliGrasp requires querying LLMs ten times to derive simple quantities, what is the total computation time for executing this process? Detailed information on computational efficiency is essential for assessing the practicality of integrating LLMs into real-time robotic operations.
* What specific improvements are observed from finetuning the LLMs on the PhysObjects dataset? Additionally, why is the GPT-3.5-Turbo model without finetuning not included in the study?
* Does the PhysObjects dataset used for finetuning the LLM include any of the objects later tested in the evaluation phase?

**Robotics Focus:**

4

**Summary Of Paper:**

The paper explores the application of large language models (LLMs) to enhance robotic grasping capabilities by leveraging their ability to reason about physical properties and generate code. It describes a novel approach where LLMs are used to infer the mass, friction coefficient, and spring constant of objects from their semantic descriptions. These physical characteristics are then translated into executable, adaptive grasp policies. The study tests these LLM-parameterized grasp policies against traditional adaptive grasp policies and direct LLM-as-code policies using a custom benchmark comprising 12 delicate and deformable items, such as food, produce, and toys, with a wide range of mass and required pick-up force. Results show that LLM-parameterized policies outperform the alternatives. Further improvements in property estimation and grasp performance are achieved through model finetuning on property-based comparisons and the use of chain-of-thought prompting to elicit these comparisons.

**Summary Of Recommendation:**

While the approach is promising and the evaluation demonstrates potential advantages, the incremental nature of the contributions and the lack of crucial comparative analysis weaken the paper’s impact. If these issues are addressed, particularly by enhancing the novelty and depth of evaluation, the paper could be a strong candidate for acceptance in future submissions. A weak reject is recommended at the current state.

---

### Official Review · Reviewer_JraE · 2024-07-22
**Manuscript review**

**Originality:** 4
**Technical Quality:** 4
**Clarity Of Presentation:** 4
**Potential Impact:** 4
**Recommendation:** 3
**Confidence:** 5

**Review:**

The authors present a promising method for utilizing LLMs' physical reasoning and code generation capabilities to infer object properties from descriptions and create grasp policies for grasping delicate objects. While the approach demonstrates clear merit, the evaluation strategy could be strengthened to more rigorously demonstrate the effectiveness of the approach and maximize the impact of this work.

**Strengths**:
- DeliGrasp demonstrates the potential of Large Language Models (LLMs) for generating grasp policies for delicate objects, an area that has been under-explored in LLM-based robotic manipulation.
- The authors present a detailed pipeline that integrates LLM-driven grasp planning with object perception, offering a practical framework for real-world application and object-related query answering.
- By integrating compliance estimation, DeliGrasp showcases the ability to extract relevant object-specific information during grasping, suggesting potential applications in various domains.
- The paper is generally well-written and easy to follow, providing a comprehensive overview of the approach and its potential contributions.


**Weakness**:
- Table 2 requires further clarification. For instance, why does the estimated mass vary for different approaches on the same object (e.g., Object IDs 8 -11) while the reported error remains consistent? A deeper explanation of these discrepancies would strengthen the analysis.
- To better assess the accuracy and limitations of the approach, including LLM-generated values for μ and k, along with ground truth mass values for objects in Table 2, would be beneficial.
- Providing examples of actual prompts and queries used with the LLM would enhance reproducibility and offer a clearer understanding of the method's implementation details.
- Figures 2 and 4 should be improved in terms of resolution and clarity to better support the paper's overall presentation.


**Other feedback**:
- Sort citations in the Related Work section
- Table 1 does not have a “grasp verb” column, though it is discussed in the text.
- Add descriptive captions to Tables 1, 2, and 3, where all symbols, columns, and rows are explained clearly. Briefly describe the salient points and main takeaways of the results/
- Consider bringing the text in lines 135-139 earlier in section 3 around lines 94-99, as it will provide a user a clearer picture of the complete pipeline.
- Please rephrase the sentences in lines 110-114 for better readability.

**Quality Of The Limitations Section:**

3

**Questions For Rebuttal:**

- Please explain why the estimated mass varies for different approaches on the same object (e.g., Object ID 11) in Table 2 while the reported error remains consistent.
- How is the Columbian friction model accounted for while generating LLM responses?
- What are the actual prompts used to generate the description and code?

**Robotics Focus:**

4

**Summary Of Paper:**

The authors introduce DeliGrasp, which utilizes LLM's physical reasoning and code generation capabilities to infer object properties from descriptions and create adaptive grasp policies. DeliGrasp tackles the challenge of grasping diverse and fragile objects like paper airplanes and ripe avocados. A complete pipeline is presented, encompassing object localization, description, and initial grasp point estimation. A dual-prompt LLM interaction generates executable grasping code. Furthermore, estimating object compliance while grasping enables answering context-specific queries about the objects for downstream tasks, such as assessing avocado ripeness for food preparation. The authors demonstrate the effectiveness of DaliGrasp by conducting multiple successful grasping experiments on a dataset of 12 diverse delicate objects.

**Summary Of Recommendation:**

The authors present a compelling approach for leveraging LLMs in the grasping of delicate objects, showcasing the potential of this methodology. However, the current evaluation strategy raises questions about the generalizability and robustness of their findings. I recommend accepting this paper only if the authors provide additional experimental analysis to address these concerns. Otherwise, I would suggest rejection due to insufficient evidence of the approach's efficacy.

---

### Author Rebuttal · Authors · 2024-08-11

We thank the reviewers for their keen and insightful comments.

Though LLM common-sense reasoning has been well leveraged in various systems, little to no work has dealt with forces and torques, which we believe to be critical for manipulation. We describe how our work is the first to deploy LLM reasoning for forceful gripper-object interaction and demonstrates novel grasping versatility and efficacy on a diverse set of delicate objects.

We have provided four new baselines based on two different classical adaptive grasp strategies from recent literature, and we observe that such methods are not as versatile or consistently successful as DeliGrasp policies. Furthermore, these classical methods require expert tuning of controller parameters.

We have clarified our prompting methodology and reorganized our methodology so that the description of our experimental setup is more cohesive. We also have improved the resolution, clarity, and details of Figures 2 and 4 and Tables 1, 2, and 3 for better readability. We then include computational expense analysis in our paper.

These additions and modifications are reflected in our individual comments and throughout the revised paper PDF attached in the ZIP file.

---

### Decision · Program_Chairs · 2024-09-04

**Decision:**

Accept

**Comment:**

Post-Rebuttal Meta Review:
-------------------------------------------
The paper presents an interesting grasping framework. Given that the rebuttal and responses addressed the major concerns of the reviewers, I am happy to recommend the paper for acceptance. We look forward to the discussion and changes being reflected in the camera-ready version.

Original Meta Review:
-------------------------------------------
This paper proposes DeliGrasp, a grasp controller whose parameter selection is based on semantic information gathered from an LLM.

Strengths: The paper proposes an interesting strategy of integrating common-sense knowledge into an adaptive grasp controller and demonstrates this on a variety of real-world objects. The paper is well-written and presents an entire system in an easy-to-understand way.

Weakness: Given the rise of systems that leverage LLMs to provide common-sense reasoning, the paper could more clearly justify how it differs from previous methods (Reviewer M14h) as well as compare to a wide range of baselines (Reviewer M14h, Reviewer uYAW). It would also strengthen the paper to clarify a few details with respect to the experimental setup (Reviewer JraE), the specification of a few figures and tables (Reviewer JraE, Reviewer uYAW) and the computational expense of the framework (Reviewer M14h).